# Facial Morphometrics in Black Celebrities: Contemporary Facial Analysis Using an Artificial Intelligence Platform

**DOI:** 10.3390/jcm12134499

**Published:** 2023-07-05

**Authors:** Cristina A. Salinas, Alice Liu, Basel A. Sharaf

**Affiliations:** Division of Plastic Surgery, Department of Surgery, Mayo Clinic, Rochester, MN 55905, USA; salinas.cristina@mayo.edu (C.A.S.); liu.alice@mayo.edu (A.L.)

**Keywords:** African American, facial aesthetics, facial morphology, facial gender affirmation, ethnic sensitive facial aesthetic surgery

## Abstract

The diversity of patients pursuing facial aesthetic and facial gender-affirming surgery (FGAS) is increasing, yet there is a paucity of objective guidelines to facilitate surgical decision-making in patients of color. We conducted a quantitative analysis of black celebrities using standardized frontal photos of 21 female and 21 male celebrities. Celebrities were chosen from popular entertainment magazines and websites, including People Magazine, the Internet Movie Database (IMDb), Cosmopolitan, and Essence. For each celebrity, 100 facial landmarks were detected through a facial analysis artificial intelligence (AI) program. Black males had greater facial height, bizygomatic width, lower facial height, and bigonial width than females. However, the facial height to bigonial width ratio was similar between genders and approximated the golden ratio (1.618). Female faces demonstrated a greater mid-face height to total facial height proportion, and males had a greater lower facial height proportion. Females exhibited an upward-slanted medial brow and shorter total eyebrow length, nose height, and alar width. Forehead height above the lateral brow was greater in males, while central forehead height was similar to females. This is the first study that has utilized AI to provide ethnicity-specific facial morphometrics relevant to facial rejuvenation and FGAS in the black population.

## 1. Introduction

The relevance of one’s race and ethnicity in the realm of plastic surgery has become increasingly important as the US population diversifies each year [1]. The accessibility and popularity of aesthetic surgery have also increased in recent years. For instance, patients of Asian, Hispanic, and African descent currently make up the fastest-growing groups that desire cosmetic surgery [2]. In 2017, the proportion of cosmetic procedures performed on non-white patients was 30% [3]. A similar trend exists in the increasing diversity of transgender patients pursuing a facial appearance that better aligns with their gender identity. For many, this journey involves facial gender-affirming surgery (FGAS), which significantly improves the mental health-related quality of life in transgender individuals [4,5]. Both facial rejuvenation surgery and FGAS require a holistic understanding of aesthetically pleasing outcomes, which are shaped by patients’ self-defined ethnicities [1,2].

Previous studies have reported significant differences in facial features among different ethnic groups [1,6,7,8]. Studying the range of facial sexual dimorphisms among different ethnicities is thus essential to providing inclusive, culturally sensitive, and evidence-based care for patients. Although facial aesthetics is well-studied in the white population, there is a paucity of objective guidelines to facilitate surgical decision-making in patients of color. This presents a significant gap, as the perception of beauty is not homogeneous but varies greatly depending on the patient’s racial and ethnic identification, gender identification, contemporary beauty trends, and most importantly, the patient’s individual preferences. Although we cannot predict the last factor, performing a focused study on contemporary aesthetic trends in patients of color can help guide clinicians and patients in their decision-making. Celebrities often exert a significant influence on society’s beauty ideals, and thus identifying patterns in their facial features may provide valuable insight into a specific ethnicity’s aesthetic standards. Here, we describe a novel method using artificial intelligence (AI) in facial recognition technology to efficiently analyze sexually dimorphic facial patterns among black celebrities. We aim to use these findings to help provide culturally sensitive, evidence-based care for African American patients undergoing FGAS or facial rejuvenation surgeries. 

## 2. Materials and Methods

Twenty-one black female and twenty-one black male celebrities were included in this study. This list was generated from celebrities featured on the cover of People magazine’s “Beautiful” issue (1999–2021), “Sexiest Man Alive” (2018–2020), the Internet Movie Database (IMDb), Cosmopolitan, and Essence. Many of the photos utilized were taken at formal entertainment events by professional photographers present at the events, based on photo credits from sources, such as Getty Images, Shutterstock, and Alamy. Images of these celebrities were used if they met the following inclusion criteria: full-face, front-view photo, fully visible facial contour, minimal facial animation, and no significant cosmetic surgery as determined by an experienced plastic surgeon. 

For each subject, 100 facial landmarks were detected through a custom, semi-automatic facial analysis program. Basic landmarks such as the eyebrows, nose, lips, medial line of the face, and facial contour were automatically detected using Vision framework, Apple’s computer vision algorithm. Additional custom points, such as the hairline points, glabella, and alar points, were added through custom-programmed MATLAB software. Each facial landmark point was manually confirmed by 3 independent graders, including the corresponding author (BAS) who has over 10 years’ experience in facial plastic surgery, to mitigate any discrepancies in photographic standards or potential digital manipulation. Pixel distances were converted to metric measurements by dividing the pixel measurement by the subject’s white-to-white corneal diameter in pixels, a method that has been previously described [9,10]. The ratio was then multiplied by the accepted mean white-to-white corneal diameter in millimeters (11.71 ± 0 0.42 mm) [9].

To validate the accuracy of this method of pixel conversion to absolute measurements, 78 facial measurements were obtained from 6 volunteers and compared against a gold standard. For each subject, frontal photos were obtained that satisfied the inclusion criteria (described above) and contained a reference ruler for scale. To obtain true physical distances, each facial landmark distance was measured manually against this reference ruler by drawing lines for each facial distance and scaling the line against the reference ruler (Figure 1). Following that, the same photos were analyzed through the AI-based method for comparison. The paired differences were tested for significance using the Wilcoxon Signed-Rank test. This average difference was found to be 1.17 ± 1.14 mm (*p*-value = 0.96). Additionally, this method of facial analysis was conducted in white celebrities for full facial analysis and hairline analysis [11,12]. 

To account for any rotation of the face, an overall face rotational angle was calculated for each photo by finding the angle between the median face vector (glabella to menton) and the vertical vector. If the angle was positive, it indicated that the face was turned clockwise relative to the vertical vector. Additionally, if a measurement had a corresponding right and left side (left and right palpebral fissure lengths), their averaged value was subsequently used in our analysis.

Analysis was performed with Blue Sky Statistics (BlueSky Statistics LLC, Version 7.40, Chicago, IL, USA), a statistical analysis software, to determine the mean, standard deviation, and P-values of the measurements and facial proportions in males and females. Values of *p* < 0.05 were considered statistically significant. The sample size calculation for this study was based on a t-test for differences in average bony mandibular width between males (9.35 ± 0.57 cm) and females (8.70 ± 0.56 cm) [13]. A study with a power of 95% would require a total sample of 18, and our study included 21 males and 21 females.

Facial height was defined as the distance from the trichion to the menton, and facial width was defined as the bizygomatic width. Forehead height was calculated from the trichion to the glabella. Forehead width was calculated as the distance between hairline points 1 and 9 (HL p1 and HL p9), which were at the same horizontal level as the glabella and the two medial eyebrow points (EB p3 and EB p4). Additionally, mid facial height was measured as the distance from the glabella to the subnasale. Lower facial height was measured as the distance from the subnasale to the menton, and jaw width was defined as the bigonial width. The ratio of horizontal facial thirds was calculated as the upper (trichion to glabella) to mid (glabella to subnasale) to lower facial height (subnasale to menton) ratio. The ratio of vertical facial thirds ratio was calculated as the right (right exocanthion to right endocanthion) to mid (right endocanthion to left endocanthion) to left (left endocanthion to left exocanthion) ratio (Figure 2).

Forehead and Hairline: Seven vertical forehead (FH) measurements (M) were taken from the hairline (HL) to the eyebrows (EB) bilaterally. The central and paramedian forehead height were defined by FH M3–M5. The lateral forehead height was defined by FH M1-2 and FH M6-7 (Figure 3).

FH M1 and FH M7 were the forehead heights above the lateral ends of the eyebrows, from HL p2 to EB p1 and from HL p8 to EB p6, respectively.FH M2 and FH M6 were the forehead heights above the eyebrow peaks, from HL p3 to EB p2 and from HL p7 to EB p5, respectively.FH M3 and FH M5 were the forehead heights above the medial ends of the eyebrows, from HL p4 to EB p3 and from HL p6 to EB p4, respectively.FH M4 was the distance from the trichion (HL p5) to the glabella. FH M4 is also the central forehead height.

Periorbital Region and Eyebrows (Figure 4)

Palpebral fissure height: highest to lowest point of the palpebral fissure.Palpebral fissure width: exocanthion to endocanthion.Interpupillary distance: right to left iris center.Eyebrow length: medial brow (from medial brow point to brow peak), lateral brow (from brow peak to lateral brow point), and total brow length (medial brow + lateral brow).

Angular Measurements (Figure 5)

Canthal tilt: The angle between a vector from the endocanthion to the exocanthion and a horizontal vector. For each subject, their face rotational angle was subtracted from the canthal angles, and the left and right canthal angles were averaged.Eyebrow angles: three angles formed from the medial brow point, brow peak, and lateral brow point were obtained. Angles were measured in relation to the interpupillary line to account for tilting of the head. Brow angle 1 is the formed by the medial brow line and a line parallel to the interpupillary line. Brow angle 2 is formed by the medial brow line and the lateral brow line. Brow angle 3 is formed by the lateral brow line and a line parallel to the interpupillary line.

Nose (Figure 4)

Alar width: ala to ala.Nose length: nasion to pronasale.

Lower Face and Jaw: Measurements were calculated from the subnasale (Sn) to 13 lower facial contour points (P1–P13) (Figure 6).

M1 and M13 are the distances from the Sn to the inferior border of the left (P1) and right earlobe (P13), respectively.M2 and M12 are the distances from the Sn to the left (P2) and right gonion (P12), respectively.M3 and M11 are the distances from the Sn to the jawline at P3 and P11, respectively. P3 and P11 are at the same vertical level as the exocanthion.M4 and M10 are the distances from the Sn to the jawline at P4 and P10, respectively. P4 and P10 are at the same vertical level as the cheilion.M5 and M9 are the distances from the Sn to the jawline at P5 and P9, respectively. P5 and P9 are at the same vertical level as the endocanthion.M6 and M8 are the distances from the Sn to the jawline at P6 and P8, respectively. P6 and P8 are at the same vertical level as the peak of cupid’s bow.M7 is the distance from the Sn to the menton (P7). M7 is also the central lower facial height.

The mean facial contour of black males and females was created using the 24 facial contour points. The male and female facial contours were overlayed to demonstrate the major differences between genders.

## 3. Results

The black female celebrities included in our study were: Adut Akech, Alek Wek, Alicia Keys, Ashanti Douglas, Beyonce Knowles, Ciara Wilson, Cynthia Erivo, Danai Gurira, Gabrielle Union, Janelle Monae, Keke Palmer, Kerry Washington, Leomie Anderson, Melissa Jefferson (Lizzo), Lupita Nyong’o, Maria Borges, Naomi Campbell, Robyn Fenty (Rihanna), Susan Watson, Tyra Banks, and Zendaya Coleman. The black male celebrities included in our study were: Adonis Bosso, Alton Mason, Boris Kodjoe, Brandon P. Bell, Broderick Hunter, Chiwetel Ejiofor, Courtney Burrell, David Agbodji, Donald Glover, Fernando Cabral, Geron McKinley, Jamie Foxx, John Legend, Karamo Brown, Michael B. Jordan, Michael Ealy, Niles Fitch, Oliver Kumbi, Omar Epps, Winston Duke, and Yahya Abdul-Mateen II.

The mean age of the females at the time of their photo was 28 (Standard deviation, SD = 5.57 years; range 18–38 years), and the mean age of the males at the time of their photo was 29 (SD = 5.01 years; range 19–37 years). Table 1 demonstrates the upper, mid, and lower facial measurements, and Table 2 demonstrates the facial proportions. Facial height and width were significantly greater in males (height = 20.40 cm, width = 14.25 cm, respectively) compared to females (height = 19.00 cm, width 13.63 cm, respectively). Forehead height and width were similar (Figure 2). Additionally, the ratio of facial height to facial width (R1) and the ratio of forehead height to forehead width (R2) were similar between males and females. Lower facial height and bigonial width were significantly greater in males by 1.13 cm and 1.10 cm, respectively (*p*-value < 0.001). The ratio of lower facial height to bigonial width (R3) was also statistically greater in males (0.59) compared to females (0.55). 

The following facial proportions were significantly greater in black males compared to black females: Facial height to forehead width (R4, Figure 7A), facial width to forehead width (R6, Figure 7B), and facial height to forehead height (R9, Figure 7C). 

Females had a greater ratio of facial width to bigonial width (R7, Figure 8A), and forehead width to bigonial width (R8, Figure 8B), signifying a more tapered facial contour from upper to lower face compared to males. 

The ratio of facial height to bigonial width (R5, Figure 9A) and total facial height to mid facial height (R10, Figure 9B) were similar between males and female.

Females adhered more closely to the ratio of horizontal and vertical facial thirds than males. Female faces exhibited mean horizontal facial thirds ratio (R11) of 1:1.09:1.06, while males exhibited a ratio of 1:1.15:1.26 (Table 3). This demonstrates that black men had a statistically longer lower facial proportion, while women had a greater midface proportion. However, the absolute midfacial height was comparable between the two genders (3.67 cm in males, 3.53 cm in females, *p*-value = 0.195). Results for the ratio of vertical facial thirds (R12) demonstrated that both males and females had a greater mid face proportion compared to the left and right (1:1.29:1 in males, 1:1.27:1 in females) (Figure 2).

### 3.1. Forehead and Hairline

Black females exhibited shorter lateral forehead height but had similar medial forehead heights compared to males (Table 4). Forehead height over the lateral brow (Avg FH M1 and FH M7) was 0.96 cm greater in males, and forehead height over the brow peak (Avg FH M2 and FH M6) was 0.99 cm greater in males. Forehead height over the medial brow (Avg FH M3 and FH M5) was slightly greater in females (6.11 cm) compared to males (6.01 cm), but the difference was insignificant (*p*-value = 0.473) (Figure 3).

### 3.2. Periorbital Region and Eyebrows

The periorbital measurements were all similar between black males and females (Table 5). Of the eyebrow lengths calculated (Table 6), the medial and lateral eyebrow lengths were comparable between the two cohorts, but the total eyebrow length was significantly longer in males (5.18 cm) compared to females (4.87 cm) (*p*-value 0.036). Additionally, the distance from the medial brow to the medial canthus was also statistically greater in males (2.38 cm) compared to females (2.35 cm) (*p*-value <0.035). However, the distance from the lateral brow to the lateral canthus was comparable between the two cohorts (Figure 4). 

### 3.3. Angular Measurements

Eyebrow angle 1 was statistically smaller in males (8.07°) compared to females (12.01°) (*p*-value < 0.03), indicating that the medial brow had a more upward slanted path in females and a more horizontal path in males. The other two eyebrow angles were similar between males and females. The canthal tilt was greater in females (7.49°) compared to males (5.93°) but was not statistically significant (Table 7, Figure 5).

### 3.4. Nasal Region

Men had a greater nose length (4.27 cm) compared to females (3.91 cm) (*p*-value = 0.031). Men also had a greater alar width (4.44 cm) than females (3.93 cm) (*p*-value < 0.001). These values are demonstrated in Table 8, and the measurements are demonstrated in Figure 4.

### 3.5. Lower Face and Jaw

In the lower face, males exhibited significantly longer lower facial measurements at all points (Table 9). Going from laterally to medially, the difference between lower facial measurements in men and women progressively increased. Compared to females, the distance from the subnasale to the bottom of the earlobe (Avg M1 and M13) was 0.43 cm greater in males. The distance from the subnasale to the gonions (Avg of M2 and M12) was 0.65 cm greater in males. Additionally, the average of M3 and M11 was 0.82 cm shorter in female faces (Figure 6).

Three lower facial measurements defined the chin. The average of M4 and M10 was 1.11 cm greater in males. The average of M5 and M9 was 1.19 cm greater in males, and this was the greatest difference observed between the lower facial measurements in males and females. The average of M6 and M8 was 1.15 cm greater in males (Figure 6). 

### 3.6. Mean Facial Contour of Black Males and Females

The major difference in the upper facial contour was that males had a greater lateral forehead height compared to the females. The facial contour was also more tapered from the lower forehead (HL p 1 and HL p9) to the gonions (P2 and P12) in females. Males had a longer lower face with more protrusion at the gonions and a wider chin at the lateral chin points, P6 and P9 (Figure 10).

## 4. Discussion

As the patient population seeking facial aesthetic surgery continues to become more diverse every year, it is important for surgeons to know how to counsel patients of different ethnic and racial backgrounds when discussing facial rejuvenation or FGAS [14]. Understanding facial morphology and facial aesthetics as perceived by the black population can increase patient satisfaction after aesthetic surgery by emphasizing ethnic preservation instead of transformation [1,15]. Farkas et al. (2007) previously reported anthropometric measurements of the average African American (AA) male and female faces [8]. However, beauty is subjective and difficult to define. Beauty standards follow trends and societal and cultural influences. This study aims at evaluating facial morphometrics in black models and celebrities to better understand the current trends in facial morphometrics. This study is not intended to define beauty standards in this population.

Facial height, facial width, lower facial height, and bigonial width were significantly greater in males compared to black female celebrities. However, forehead height and width were comparable between the two. The ratio of facial height to facial width (R1) was 1.43 in black male celebrities, compared to 1.39 in the historic average AA male [8]. In females, this ratio was 1.40 and similar to the historic ratio of 1.38 in average AA female [8]. Interestingly, the ratio of lower facial height to bigonial width (R3) was distinctly smaller in both black male and female celebrities, compared to historic ratios in the average AA population (R3 = 0.59 in black male celebrities, compared to 0.76 in the average AA male; 0.55 in black female celebrities, compared to 0.74 in the average AA female) [8]. Additionally, the ratio of facial height to bigonial width (R5, Figure 9A) is similar between black male and female celebrities and approximated the golden ratio of 1.618 (1.61 and 1.64, respectively). However, this ratio in the average AA population was 1.85 in males and 1.86 in females. These results indicate that black faces portrayed as aesthetically pleasing in popular media have a greater bigonial width compared to the general AA population.

Analysis of the ratio of horizontal facial thirds in black celebrities demonstrated that females adhered more closely to the 1:1:1 ratio compared to males who exhibited larger mid and lower facial proportions. Both male and female celebrities deviated from a perfect ratio of vertical thirds and exhibited a larger intercanthal distance compared to faces of the average white adult male or female. The average exocanthion to endocanthion measurement (palpebral fissure width) was similar between black males and females (2.86 cm and 2.79 cm, respectively).

A previous study evaluated AA profiles in fashion magazines and how they have evolved from 1940 to the 1990s. They found that the upper face stayed constant while the mid and lower face had changed. In the later years, models tended to have fuller and more anteriorly positioned lips and a more acute nasolabial angle [16]. The changes reflected the facial preferences of AA individuals when surveyed on which facial structures they found most aesthetically pleasant, which included the lips, chin, and nose [15,17]. 

Rhinoplasty in the non-white patient population was previously influenced by the ideal white nose. Contemporary ethnic rhinoplasty, however, has evolved to preserve the harmony of ethnic facial features. The nasal anatomy of black individuals has been described as having shorter and flatter nasal bones, a broad middle vault and alar base, strong ala, and a bulbous or amorphous tip [1]. The most common nasal areas of desired modification in the black population are the nasal tip, alar base, and nostril size [18]. Our analysis of the frontal view of the nose demonstrated that black males had significantly longer noses and wider alar bases than black females. The length of the nose was 4.27 cm in males and 3.91 cm in females (*p*-value = 0.031), while alar width was 4.44 cm in males and 3.93 cm in females (*p*-value < 0.001). Our results corroborate with previously reported values of mean alar width in average AA male and female faces (4.31 cm and 4.01 cm, respectively) [8,19]. 

In FGAS, foreheadplasty is a common procedure performed to effectively feminize or masculinize a face [20,21,22,23]. The central (FH M4) and paramedian forehead height (over the medial eyebrow) were similar between black male and female celebrities. However, the lateral forehead height above the brow peak (Avg FH M2 and FH M6) and the lateral brow (Avg FH M1 and FH M7) were significantly greater in the male celebrities (4.93 cm and 4.18 cm, respectively) compared to female celebrities (3.94 cm and 3.22 cm, respectively), indicating significant sexual dimorphism in lateral forehead height. These results are important when considering hairline manipulation and choosing a method of hairline advancement. 

Similarly, mandibular manipulation greatly contributes to the perceived gender of a face, with surgical options including genioplasty, chin augmentation, mandibular contouring, and neck lift or liposuction [24,25,26]. Black males had greater lower facial measurements at all points compared to black females. Going laterally (measurement from the subnasale to the bottom of the earlobe) to medially (measurement of the subnasale to the menton), the difference between the lower facial measurements of males and females progressively increased, with the greatest difference observed at the lateral chin points (Avg M5 and M9). These results, as depicted in the facial contours illustration, demonstrate that males have longer and wider chins compared to females, who exhibited a narrower and more tapered chin contour (Figure 10).

Previous studies have compared facial morphometrics of the average adult AA and white populations. Compared to white females, AA females have greater facial width, a wider alar base, greater width from cheilion to cheilion, more protruded lips, smaller ear length, and a greater palpebral fissure width (exocanthion to endocanthion). White females have a more prominent chin and malar region and a shorter lower facial height compared to AA females. [27,28]. In a study investigating facial changes from adolescence to adulthood, AA female faces experienced an increase in length and width of the face, an increase in nasal tip projection, and a decrease in the periorbital region. White female faces experienced an increase in nose and chin projection going from adolescence to adulthood [29]. Compared to white males, AA males have more prominent upper foreheads and supraorbital ridges, a shorter midface proportion, greater alar width, a more acute nasolabial angle, and more protruded lips. White males have a more prominent nasal tip and malar region compared to AA males, however there is little difference in the chin [19,27,28]. Facial aging also differs between black and white faces. Black faces experience significantly less glabellar angle decrease (0.68°) compared to white faces (2.3°) from ages 47 to 58. Additionally, the mean maxillary angles decreased significantly in white faces, while there was no change in black faces over time [30,31]. Within the black population studied, females experienced a significant increase in orbital width, while there was no significant change in the males [31].

While the African American population shares a similarity in African descent, individual population and racial differences contribute to unique facial morphology within this larger population group. Several studies have shown differences in facial dimorphism of individuals from different African countries. Compared to AA individuals, Kenyan males exhibited greater lower facial height and intercanthal distance, while Zimbabwean males had more prominent zygomas and lateral supraorbital region. On the other hand, AA females had more pronounced malar regions, lips, forehead, and lateral perioral regions compared to Zimbabwean females [32,33]. Additionally, a comparison of Moroccan and Senegalese adults revealed that Moroccans had longer and more anteriorly positioned noses and a more prominent chin in the profile view [34].

There are a few limitations to our study. First, the photos included were obtained from online sources and not photographed specifically for this study, which could introduce variation in camera shooting angle, facial tilt, and rotation. However, every photograph used was carefully evaluated for inclusion in our analysis. Most photos were taken at major entertainment events by professional photographers based on the photo credits of photo sources, such as Shutterstock, Getty Images, and Alamy. All photos were then reviewed by three independent graders and were only included only if they met the criteria of a front-view photo, visible facial contour, and minimal animation. Another limitation is that we cannot ensure that there was no surgical or digital alteration, neuromodulation, surgical, or non-surgical alteration of the face. We evaluated multiple images of the same celebrity to assess potential for cosmetic surgery or digital alteration. However, despite potential smaller alterations that would change the celebrity’s native facial appearance, these changes were likely made to enhance the subject’s appearance, and thus would still reflect the current idea of an aesthetically pleasing, black face. Finally, we did not distinguish between black celebrities of different racial descent. We chose to include this sample of celebrities based on popularity in the United States and relevance in the entertainment industry.

## 5. Conclusions

Analysis of black celebrity faces demonstrated that males had longer faces and a wider mid and lower face compared to females. Male faces had longer total eyebrow lengths, a more horizontal tilt of the medial brow, greater nasal length, and wider alar bases. The ratio of facial height to bigonial width in both black male and female celebrities approximated the golden ratio. Females had significantly shorter lateral forehead height compared to men and a more tapered contour from the upper to lower face. Men exhibited a wider lower face and chin contour. This analysis provides important insight into ethnicity-specific facial morphometrics relevant to surgical and nonsurgical facial rejuvenation. Furthermore, this study adds important parameters when planning facial gender-affirming surgery in the African American population. 

## Figures and Tables

**Figure 1 jcm-12-04499-f001:**
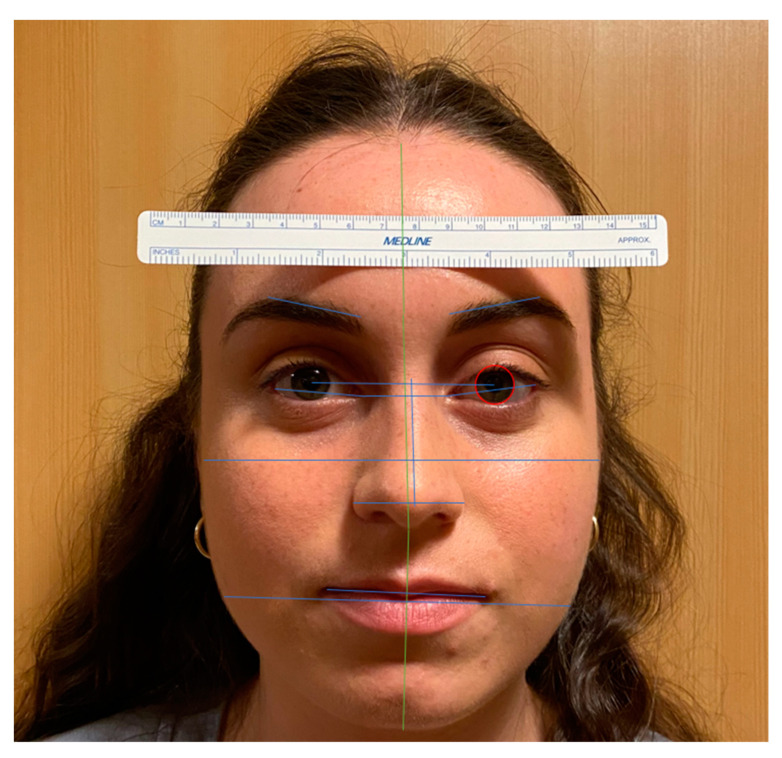
Demonstration of the facial measurements obtained on a volunteer’s face to validate our method of pixel conversion to absolute measurements.

**Figure 2 jcm-12-04499-f002:**
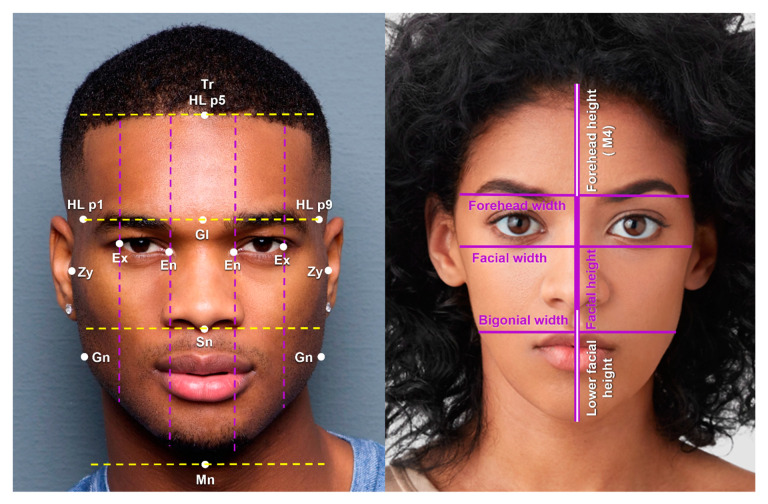
The main facial landmark points, measurements, and proportions. Facial height (trichion to menton), facial width (zygoma to zygoma), forehead width (hairline point 1 to hairline point 9), and bigonial width (gonion to gonion) are demonstrated by the solid purple lines. Forehead height (M4) and lower facial height are demonstrated by the solid white lines. Vertical facial thirds are demonstrated by the dashed purple lines, from exocanthion to endocanthion, endocanthion to endocanthion, and endocanthion to exocanthion. Horizontal facial thirds are demonstrated by the dashed yellow lines. Photos used with permission from Shutterstock. (Abbreviations: Tr, Trichion; HL, hairline; Gl, Glabella; En, endocanthion; Ex, exocanthion; Zy, zygoma; Gn, gonion; Sn, subnasale; Mn, menton; M, measurement; p, point).

**Figure 3 jcm-12-04499-f003:**
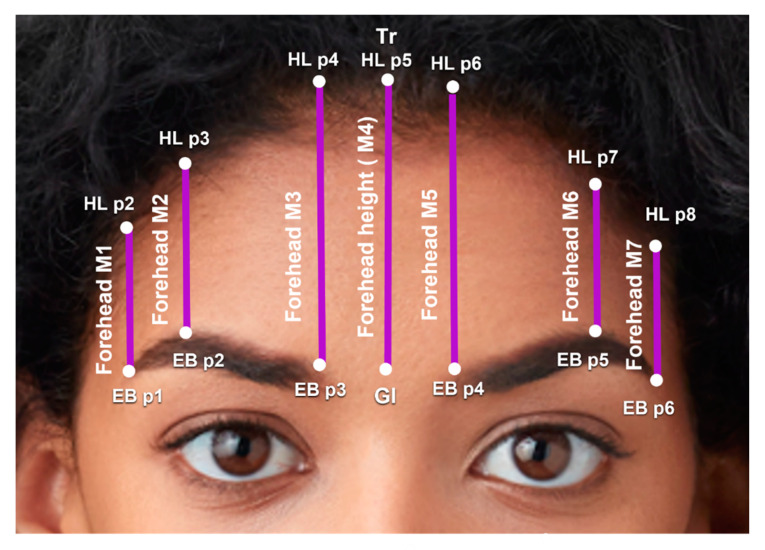
The seven forehead measurements from the eyebrows (EB) to the hairline (HL). Forehead measurement 1 (FH M1) and FH M7 are the forehead heights above the lateral ends of the eyebrows. FH M2 and FH M6 are the forehead heights above brow peaks. FH M3 and FH M5 are the forehead heights above the medial ends of the eyebrows. FH M4 is the distance from the trichion (Tr, HL p5) to the glabella (Gl) and the central forehead height. Photo used with permission from Shutterstock.

**Figure 4 jcm-12-04499-f004:**
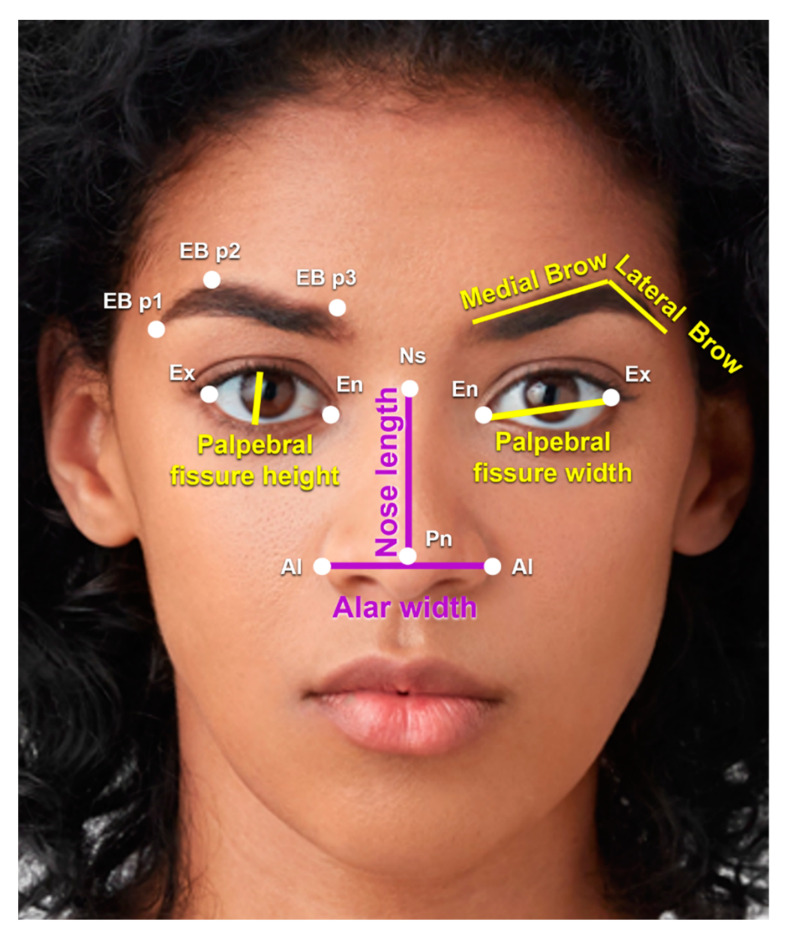
Periorbital measurements. Palpebral fissure height is the distance from the highest to lowest point of the palpebral fissure. Palpebral fissure width is the distance from the endocanthion (En) to exocanthion (Ex). The medial brow length is measured from the medial brow point (EB p3) to the brow peak (EB p2). The lateral brow length is measured from the brow peak (EB p2) to the lateral brow point (EB p1). Total brow length is the length of the medial and lateral brow. Interpupillary distance is measured from one endocanthion to the other. Nose length is the distance from the nasion (Ns) to the pronasale (Pn), and alar width is the distance from ala (Al) to ala. Photo used with permission from Shutterstock.

**Figure 5 jcm-12-04499-f005:**
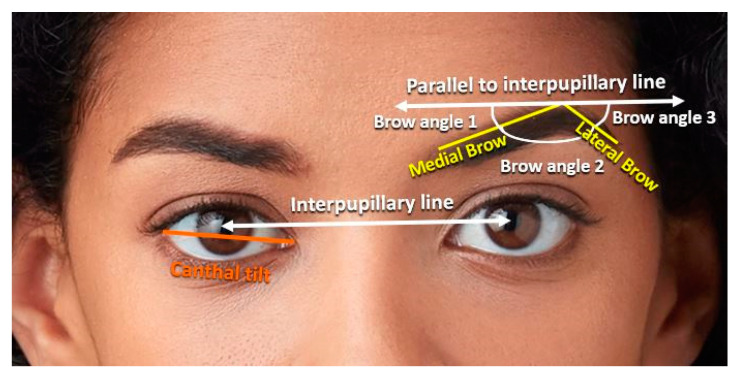
The three eyebrow angles. Angles were measured in relation to the interpupillary line to account for tilting of the head. Brow angle 1 is formed by the medial brow line (yellow) and a line parallel to the interpupillary line (white). Brow angle 2 is formed by the medial brow line and the lateral brow line (yellow). Brow angle 3 is formed by the lateral brow line and a line parallel to the interpupillary line. Canthal tilt is demonstrated by the orange line. Photos used with permission from Shutterstock.

**Figure 6 jcm-12-04499-f006:**
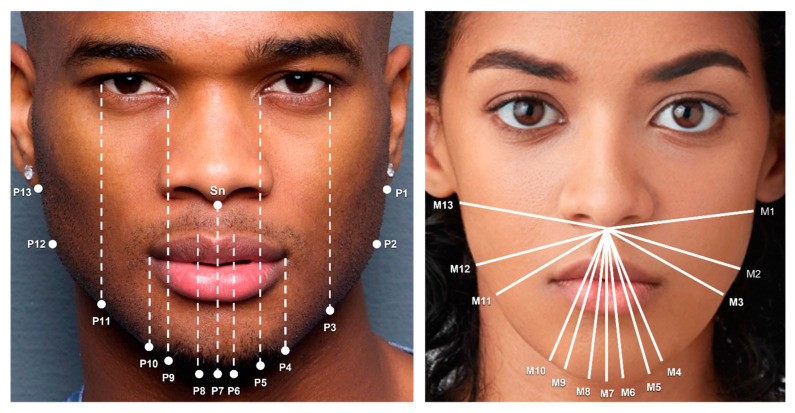
The 13 lower facial measurements. M1 and M13 are the distances from the subnasale (Sn) to the inferior border of the left (P1) and right earlobe (P13), respectively. M2 and M12 are the distances from the Sn to the left (P2) and right gonion (P12), respectively. M3 and M11 are the distances from the Sn to the jawline at P3 and P11, respectively. M4 and M10 are the distances from the Sn to the jawline at P4 and P10, respectively. M5 and M9 are the distances from the Sn to the jawline at P5 and P9, respectively. M6 and M8 are the distances from the Sn to the jawline at P6 and P8, respectively. M7 is the distance from the Sn to the menton (P7) and the central lower facial height. Photos used with permission from Shutterstock.

**Figure 7 jcm-12-04499-f007:**
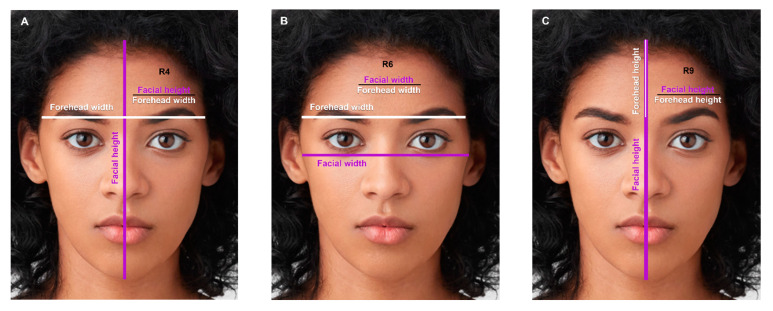
(**A**) The ratio of facial height to forehead width (R4): 3.45 in males, 3.17 in females (*p*-value = 0.011). (**B**) The ratio of facial width to forehead width (R6): 1.04 in males, 1.01 in females (*p*-value = 0.002). (**C**) The ratio of facial height to forehead height (R9): 3.45 in males, 3.17 in females (*p*-value = 0.011). Photos used with permission from Shutterstock.

**Figure 8 jcm-12-04499-f008:**
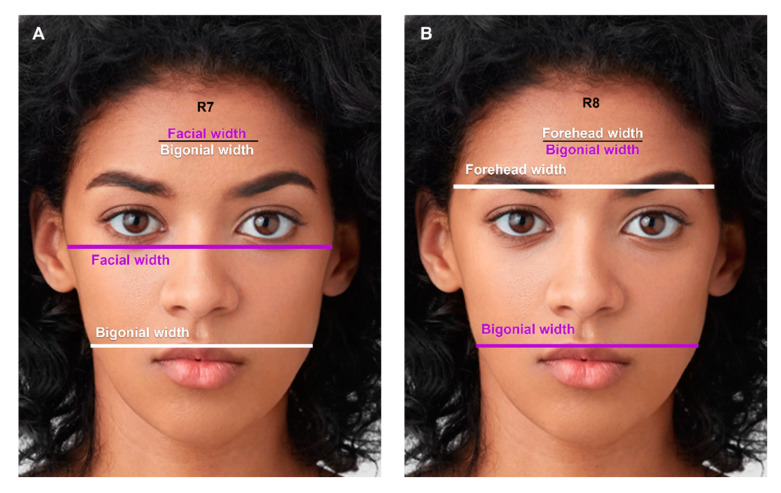
(**A**) The ratio of facial width to bigonial width (R7): 1.12 in males, 1.18 in females (*p*-value < 0.001). (**B**) The ratio of forehead width to bigonial width (R8): 1.08 in males, 1.17 in females (*p*-value < 0.001). Photos used with permission from Shutterstock.

**Figure 9 jcm-12-04499-f009:**
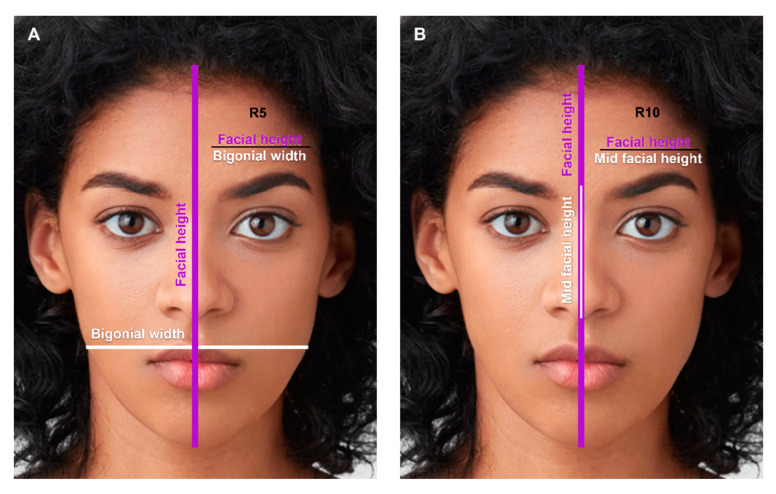
(**A**) The ratio of facial height to bigonial width (R5): 1.61 in males, 1.64 in females (*p*-value = 0.163). (**B**) The ratio of total facial height to mid facial height (R10): 2.97 in males, 2.90 in females (*p*-value = 0.297). Photos used with permission from Shutterstock.

**Figure 10 jcm-12-04499-f010:**
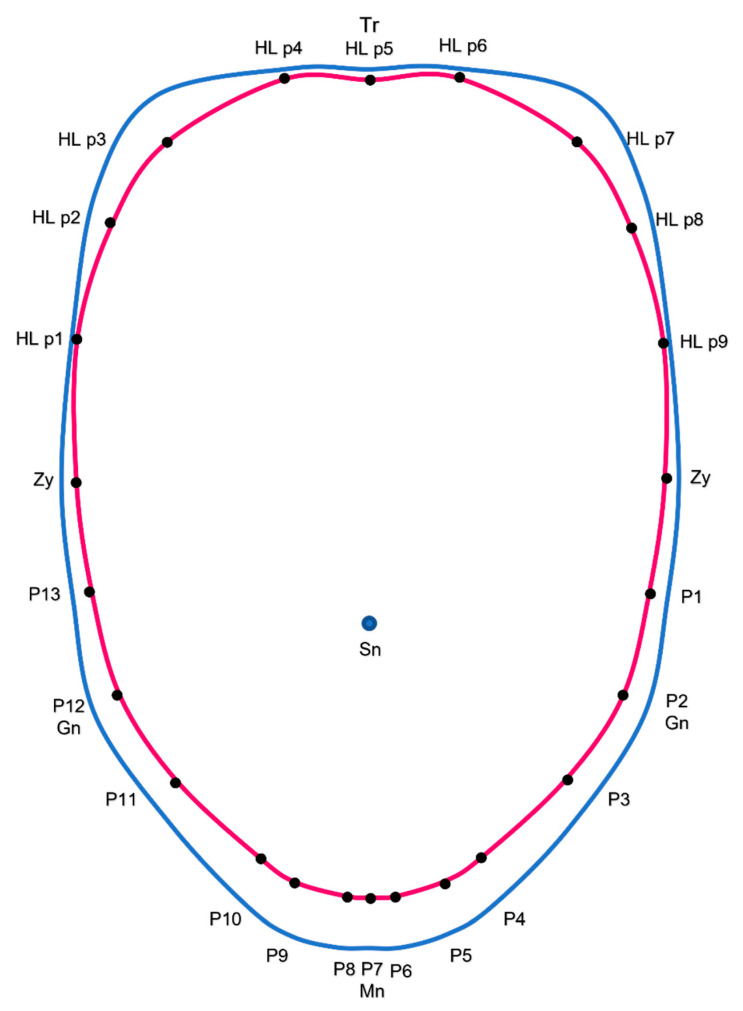
The mean facial contours of the male (blue) and female (pink) black celebrities. The subnasale is demonstrated by the blue circle in the center. Abbreviations: Tr, trichion; HL, hairline; Zy, zygoma; Gn, gonion; Mn, menton; p, point; Sn, subnasale.

**Table 1 jcm-12-04499-t001:** Upper, middle, and lower facial measurements in black celebrities.

Measurements	Points	Females, cm (n = 21)	Males, cm (n = 21)	*p*-Value
Facial Height	Trichion-Menton	19.00 (±0.99)	20.40 (±1.75)	0.002 *
Facial Width	Zygoma-Zygoma	13.63 (±0.83)	14.25 (±0.99)	0.046 *
Forehead Height	Trichion (HL p5)-Glabella	6.03 (±0.61)	5.98 (±0.89)	0.505
Forehead Width	HL p1-HL p9	13.56 (±0.82)	13.73 (±1.01)	0.753
Mid Facial height	Glabella-Subnasale	3.53 (±0.29)	3.67 (±0.27)	0.195
Lower Facial Height	Subnasale-Menton (P7)	6.40 (±0.52)	7.53 (±0.79)	<0.001 *
Bigonial Width	Gonion (P2)-Gonion (P12)	11.61 (±0.89)	12.71 (±0.98)	<0.001 *

* significant value, *p* < 0.05. Abbreviations: HL = hairline, p = point.

**Table 2 jcm-12-04499-t002:** Facial proportions of black celebrities.

Ratios	Measurements	Females (n = 21)	Males (n = 21)	*p*-Value
R1	Facial Height: Facial Width	1.40 (±0.068)	1.43 (±0.087)	0.105
R2	Forehead Height: Forehead Width	0.45 (±0.05)	0.44 (±0.06)	0.443
R3	Lower Facial Height: Bigonial Width	0.55 (±0.04)	0.59 (±0.04)	0.03 *
R4	Facial Height: Forehead Width	3.17 (±0.26)	3.45 (±0.38)	0.011 *
R5	Facial Height: Bigonial Width	1.64 (±0.10)	1.61 (±0.08)	0.163
R6	Facial Width: Forehead Width	1.01 (±0.03)	1.04 (±0.03)	0.002 *
R7	Facial Width: Bigonial Width	1.18 (±0.04)	1.12 (±0.05)	<0.001 *
R8	Forehead Width: Bigonial Width	1.17 (±0.06)	1.08 (±0.05)	<0.001 *
R9	Facial Height: Forehead Height	3.17 (±0.26)	3.45 (±0.38)	0.011 *
R10	Facial Height: Mid Facial Height	2.90 (±0.16)	2.97 (±0.21)	0.297

* significant value, *p* < 0.05.

**Table 3 jcm-12-04499-t003:** Horizontal and vertical facial thirds ratios.

Ratios	Definition	Measurements	Females (n = 21)	Males (n = 21)
R11	Horizontal Facial Thirds	Upper:Mid:Lower	1:1.09:1.06	1:1.15:1.26
R12	Vertical Facial Thirds	Right:Mid:Left	1:1.27:1	1:1.29:1

**Table 4 jcm-12-04499-t004:** Forehead height measurements.

Measurements	Points	Females, cm (n = 21)	Males, cm (n = 21)	*p*-Value
FH M1	HL p1-HL p9	3.20 (±0.81)	4.16 (±1.41)	0.023 *
FH M2	HL p2-EB p1	3.96 (±0.68)	5.02 (±0.84)	<0.001 *
FH M3	HL p3-EB p2	6.10 (±0.55)	6.07 (±0.89)	0.660
FH M4	HL p4-EB p3	6.03 (±0.61)	5.98 (±0.89)	0.505
FH M5	HL p6-EB p4	6.12 (±0.54)	5.96 (±0.91)	0.333
FH M6	HL p7-EB p5	3.91 (±0.72)	4.84 (±0.82)	0.001 *
FH M7	HL p8-EB p6	3.23 (±0.79)	4.21 (±1.44)	0.028 *
Avg FH M1 and FH M7		3.22 (±0.79)	4.18 (±1.42)	0.034 *
Avg FH M2 and FH M6		3.94 (±0.70)	4.93 (±0.82)	<0.001 *
Avg FH M3 and FH M5		6.11 (±0.54)	6.01 (±0.90)	0.473

* significant value, *p* < 0.05. Abbreviation: FH = forehead. M = measurement. HL = hairline. p = point. Avg = average.

**Table 5 jcm-12-04499-t005:** Periorbital measurements.

Measurements	Points	Females, cm (n = 21)	Males, cm (n = 21)	*p*-Value
Left Palpebral Fissure Height	Highest to Lowest Point of Palpebral Fissure	0.96 (±0.09)	0.95 (±0.10)	0.554
Right Palpebral Fissure Height		0.95 (±0.09)	0.97 (±0.10)	0.990
Avg Palpebral Fissure Height		0.03 (±0.02)	0.03 (±0.02)	0.715
Left Palpebral Fissure Width	Exocanthion–Endocanthion	2.79 (±0.14)	2.86 (±0.24)	0.333
Right Palpebral Fissure Width		2.79 (±0.14)	2.86 (±0.23)	0.642
Avg Palpebral Fissure Width		2.79 (±0.13)	2.86 (±0.23)	0.505
Interpupillary Distance	Right to Left Iris Center	6.59 (±0.39)	6.75 (±0.48)	0.399

**Table 6 jcm-12-04499-t006:** Eyebrow length measurements.

Measurements	Points	Females, cm (n = 21)	Males, cm (n = 21)	*p*-Value
Left Medial Eyebrow Length	Medial Eyebrow (EB p4)–Brow Peak (EB p5)	3.11 (±0.28)	3.28 (±0.42)	0.263
Right Medial Eyebrow Length	EB p3–EB P2	3.16 (±0.29)	3.34 (±0.33)	0.068
Avg Medial Eyebrow Length		3.13 (±0.25)	3.31 (±0.35)	0.141
Left Lateral Eyebrow Length	Brow Peak (EB p5)–Lateral Eyebrow (EB p6)	1.73 (±0.26)	1.89 (±0.32)	0.122
Right Lateral Eyebrow Length	EB p2–EB p1	1.73 (±0.26)	1.85 (±0.40)	0.170
Avg Lateral Eyebrow Length		1.73 (±0.23)	1.87 (±0.31)	0.128
Left Total Eyebrow Length	Sum of Medial and Lateral Eyebrow Lengths	4.85 (±0.37)	5.17 (±0.59)	0.080
Right Total Eyebrow Length		4.88 (±0.39)	5.19 (±0.49)	0.048 *
Avg Total Eyebrow Length		4.87 (±0.35)	5.18 (±0.52)	0.036 *
Lateral Brow to Lateral Canthus	Average of EB p1-Ex and EB p6-Ex	2.02 (±0.22)	1.87 (±0.21)	<0.819
Medial Brow to Medial Canthus	Average of EB p3-En and EB p4-En	2.35 (±0.29)	2.38 (±0.58)	<0.035 *

* significant value, *p* < 0.05. Abbreviations: EB = eyebrow. Ex = Exocanthion. En = Endocanthion. p = point. Avg = average.

**Table 7 jcm-12-04499-t007:** Eyebrow angles and canthal tilt.

Measurements	Females, Degrees (n = 21)	Males, degrees (n = 21)	*p*-Value
Brow Angle 1	12.01 (±4.99)	8.07 (±6.39)	<0.03 *
Brow Angle 2	54.99 (±8.12)	55.32 (±6.38)	<0.852
Brow Angle 3	42.8 (±8.84)	47.25 (±6.38)	<0.112
Canthal Tilt	7.49 (±2.86)	5.93 (±2.27)	<0.059

* significant value, *p* < 0.05.

**Table 8 jcm-12-04499-t008:** Measurements of the nasal region.

Measurements	Points	Females, cm (n = 21)	Males, cm (n = 21)	*p*-Value
Nose Length	Nasion-Pronasale	3.91 (±0.44)	4.27 (±0.47)	0.031 *
Alar Width	Ala-Ala	3.93 (±0.41)	4.44 (±0.64)	<0.001 *

* significant value, *p* < 0.05.

**Table 9 jcm-12-04499-t009:** Lower facial measurements.

Measurements	Points	Females, cm (n = 21)	Males, cm (n = 21)	*p*-Value
Lower Facial Measurement (M1)	Sn-P1	6.40 (±0.55)	6.78 (±0.53)	0.046 *
Lower Facial Measurement (M2)	Sn-P2	5.96 (±0.60)	6.59 (±0.59)	0.003 *
Lower Facial Measurement (M3)	Sn-P3	5.73 (±0.57)	6.56 (±0.66)	<0.001 *
Lower Facial Measurement (M4)	Sn-P4	6.01 (±0.56)	7.12 (±0.76)	<0.001 *
Lower Facial Measurement (M5)	Sn-P5	6.25 (±0.52)	7.45 (±0.77)	<0.001 *
Lower Facial Measurement (M6)	Sn-P6	6.37 (±0.50)	7.53 (±0.79)	<0.001 *
Lower Facial Measurement (M7)	Sn-P7	6.40 (±0.52)	7.53 (±0.79)	<0.001 *
Lower Facial Measurement (M8)	Sn-P8	6.40 (±0.50)	7.55 (±0.81)	<0.001 *
Lower Facial Measurement (M9)	Sn-P9	6.31 (±0.51)	7.49 (±0.82)	<0.001 *
Lower Facial Measurement (M10)	Sn-P10	6.09 (±0.53)	7.20 (±0.79)	<0.001 *
Lower Facial Measurement (M11)	Sn-P11	5.60 (±0.55)	6.70 (±0.73)	<0.001 *
Lower Facial Measurement (M12)	Sn-P12	6.19 (±0.45)	6.87 (±0.67)	<0.001 *
Lower Facial Measurement (M13)	Sn-P13	6.59 (±0.47)	7.06 (±0.60)	0.011 *
Avg M1 and M13		6.49 (±0.45)	6.92 (±0.51)	0.014 *
Avg M2 and M12		6.08 (±0.48)	6.73 (±0.59)	<0.001 *
Avg M3 and M11		5.81 (±0.53)	6.63 (±0.68)	<0.001 *
Avg M4 and M10		6.05 (±0.54)	7.16 (±0.77)	<0.001 *
Avg M5 and M9		6.28 (±0.51)	7.47 (±0.79)	<0.001 *
Avg M6 and M8		6.39 (±0.50)	7.54 (±0.80)	<0.001 *

* Significant value, *p* < 0.05. Abbreviations: Sn = Subnasale. M = Measurement. P = point.

## Data Availability

Data available on request due to restrictions e.g., privacy or ethical.

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
