# Peer review of "Facial Morphometrics in Black Celebrities: Contemporary Facial Analysis Using an Artificial Intelligence Platform"

_jcm, 2023, doi:10.3390/jcm12134499_

Round 1

Reviewer 1 Report

The authors do a very good job addressing an important question for facial plastic and plastic surgeons; what are the "ideal" facial morphometrics in black male and female faces. These results are most applicable to gender affirmation surgery. However, they are also helpful in facial rejuvenation surgery when trying to restore a youthful appearance in black patients. There are some concerns I have that need to be addressed: 

1) The term African American is used throughout the paper. A more appropriate term is black since at least two (Adut Akech, Cynthia Erivo) of the celebrities studied are not "African American." While the authors correctly point out in the discussion morphometric differences between different black populations, black is still a more appropriate term. It also makes this paper less American-centric, important for the international audience that JCM has. 

2) These measurements are obtained from young adult black males and females. This should briefly be discussed in the discussion as another important question that is not answered by this research is how to black faces age? Is it different from white populations? 

3) My suspicion is many of the faces studied had Botox given the frequency with which celebrities use Botox and these photos were taken at big events. This should be added to the limitation section as Botox would potentially impact the brow, forehead, and peri-orbital measurements. 

4) One of the main motivations for this paper is that most literature we use in facial rejuvenation and gender affirmation surgery is based on white populations. Please include more discussion comparing how male and female faces differ in white versus black populations. 

5) All measurements need to have units (see line 98)

6) Culture is misused somewhat in this paper (e.g., line 333). Skin color is only one of many cultural determinants. 

Author Response

Reviewer 1:

The authors do a very good job addressing an important question for facial plastic and plastic surgeons; what are the "ideal" facial morphometrics in black male and female faces. These results are most applicable to gender affirmation surgery. However, they are also helpful in facial rejuvenation surgery when trying to restore a youthful appearance in black patients. There are some concerns I have that need to be addressed: 

  • The term African American is used throughout the paper. A more appropriate term is black since at least two (Adut Akech, Cynthia Erivo) of the celebrities studied are not "African American." While the authors correctly point out in the discussion morphometric differences between different black populations, black is still a more appropriate term. It also makes this paper less American-centric, important for the international audience that JCM has. 
    1. Response: Thank you for your thoughtful comment. We have changed the term “African American” to “black” when the term is more appropriate and when referring to the celebrities included in this study. We have kept the term “African American” when the article cited utilized this term in their paper.

  • These measurements are obtained from young adult black males and females. This should briefly be discussed in the discussion as another important question that is not answered by this research is how to black faces age? Is it different from white populations? 
    1. Response: We have included a brief discussion of how facial aging differs between the white and black population in the discussion, starting at line 407.

  • My suspicion is many of the faces studied had Botox given the frequency with which celebrities use Botox and these photos were taken at big events. This should be added to the limitation section as Botox would potentially impact the brow, forehead, and peri-orbital measurements. 
    1. Response: We have added to the limitations sections, stating that we cannot ensure that no surgical or nonsurgical (including Botox) alteration were performed on the face. However, any changes to the native facial appearance would have been done to reflect the current idea of a beautiful face.

  • One of the main motivations for this paper is that most literature we use in facial rejuvenation and gender affirmation surgery is based on white populations. Please include more discussion comparing how male and female faces differ in white versus black populations. 
    1. Response: This has been expanded on in the discussion section, starting at line 398.

  • All measurements need to have units (see line 98)
    1. Response: the units have been added.

  • Culture is misused somewhat in this paper (e.g., line 333). Skin color is only one of many cultural determinants. 
    1. Response: The term “cultural” preservation/transformation has been changed to “ethnic preservation/transformation.”

Reviewer 2 Report

Although it is shown in limitation, the most important thing in AI analysis is an accurate data sample, in this paper, an accurately taken picture. However, as the author recognized, the analysis performed on photographs, not the purpose of accurate facial measurement, has too many artifacts and errors in the data sample itself, so the sample setting seems to be completely wrong. It is a novel attempt, but it seems difficult to draw scientific results with the current concept of data analysis. In the future, if the method of big data analysis using 20,000 to 30,000 digital images on the Internet is used, this concept analysis will be possible.

Author Response

Reviewer 2:

Although it is shown in limitation, the most important thing in AI analysis is an accurate data sample, in this paper, an accurately taken picture. However, as the author recognized, the analysis performed on photographs, not the purpose of accurate facial measurement, has too many artifacts and errors in the data sample itself, so the sample setting seems to be completely wrong. It is a novel attempt, but it seems difficult to draw scientific results with the current concept of data analysis. In the future, if the method of big data analysis using 20,000 to 30,000 digital images on the Internet is used, this concept analysis will be possible.

Response: Thank you for your review and your comment. We respectfully disagree with the general comment made that all our data is inaccurate.

All photographs selected were carefully chosen according to strict inclusion criterion with the input of the senior author (BAS). To be included, a photograph had to be: full-face, front-view photo, fully visible facial contour, minimal facial animation, and no significant cosmetic surgery as determined by an experienced plastic surgeon. The majority of the photographs selected had the photographer’s name in their credits and were taken at major entertainment and media events and Galas to lessen the odds of digital morphing. Also, each photograph selected was compared to multiple other photographs of the same celebrity to ensure consistency. While we cannot rule out potential digital alterations of the photos, we believe that the steps taken ensure improved accuracy and consistency.

To address your concern, we performed a validation of our method of measuring facial landmarks: we measured 78 facial measurements taken from 6 volunteers using this method and compared them to manual measurements taken from the subjects’ photos containing a reference ruler for scale. The mean difference between the 2 measurements was found to be small: 1.17 +/- 1.14 mm.

Lastly, this analysis was conducted in white celebrities using the same methods outlined in this current manuscript for full facial analysis and for hairline analysis. Both studies have been peer reviewed and accepted for publication at these top tier journals in plastic surgery (Plastic and Reconstructive Surgery Global Open and Aesthetic Surgery Open Forum).

We have included these recent citations in this manuscript and hope they will support the validity of our current methods (line 92). This explanation has also been added to the methods section, line 87.

Round 2

Reviewer 1 Report

The authors have done a nice job making the terminology more appropriate (i.e., black instead of African American, ethnic instead of culture). This is a nice paper. Few minor revisions: 

1) African American should be Black in the following: title; last sentence of abstract; line 69

2) Line 63, "minority races" should say "patients of color." Minority is a term that depends on ethnic makeup of a region and this is an international journal

Author Response

-

Reviewer 2 Report

As you well know, PRS is a top-tier SCI journal, but PRS Global Open is an emerging SCI journal that has not yet been recognized as SCI. In any case, it would be better for the author to explain the exact method of validation using 78 points accepted in two journals in the journal with a picture. And while the analysis of celebrities in the title is correct, this does not represent beauty, so we need to adjust the title more carefully  

Author Response

-
